# Knowledge and Visual Differentiation Ability of the Pressure Injury Classification System and Incontinence-Associated Dermatitis among Hospital Nurses: A Descriptive Study

**DOI:** 10.3390/healthcare12020145

**Published:** 2024-01-08

**Authors:** Seungmi Park, Eun Jung Kim, Son Ja Lee, Eun Jeong Kim, Ji Yeon Lee, Jung Eun Hong

**Affiliations:** 1Department of Nursing Science, Chungbuk National University, Cheongju 28644, Chungbuk, Republic of Korea; spark2020@chungbuk.ac.kr; 2Department of Nursing, Chungbuk National University Hospital, Cheongju 28644, Chungbuk, Republic of Korea; kejpsy@cbnuh.or.kr; 3Department of Nursing Administration, Chungbuk National University Hospital, Cheongju 28644, Chungbuk, Republic of Korea; 4Quality Improvement Team, Chungbuk National University Hospital, Cheongju 28644, Chungbuk, Republic of Korea; 5Department of Nursing, Seojeong University, Yangju 11429, Gyeonggi-do, Republic of Korea

**Keywords:** clinical nurses, visual differentiation, diagnostic ability, pressure injury, incontinence-associated dermatitis

## Abstract

This study investigated clinical nurses’ knowledge and visual differentiation ability of the pressure injury classification system (PICS) and incontinence-associated dermatitis (IAD), additionally analyzing possible influencing factors. A convenience sample of 248 nurses took the PICS and IAD knowledge test (KT) and completed the visual differentiation ability test (VDAT), consisting of 21 photographs with clinical information. The overall mean score for correct answers was 12.65 ± 2.90 points in PICS and IAD KT and 11.43 ± 4.57 points in VDAT. Incorrect responses were most common for statements related to stage II, III, IAD for PICS and IAD KT, and deep tissue pressure injury (DTPI), unstageable, and stage III for VDAT. Significant correlations were found between PICS and IAD KT and VDAT (r = 0.252, *p* < 0.001). Factors affecting scores for VDAT were the scores of PICS and IAD KT, debridement experience in nursing patients with PI, and the management frequency of PI and IAD. Results indicate that nurses have an overall understanding of PICS and IAD, but low visual differentiation ability regarding stage III, DTPI, and unstageable PI. Continuing education is needed to further improve knowledge and visual differentiation ability for PICS and IAD.

## 1. Introduction

Localized damage to the skin and underlying tissues at the site of bony prominence due to sustained pressure or shear force has been referred to as bed sores, pressure sores, and pressure ulcers, with the most recent term being pressure injury (PI) [1]. The number of PI patients in Korea increased by 9.1% from 23,354 in 2017 to 25,843 in 2021 [2]. The prevalence of PI varies widely depending on the type of medical institution and patient population, with the prevalence of PI ranging from 3.9–39.0% in general wards and 14.9–33.0% in intensive care units in foreign countries [3,4] to 9.7–12.0% in general wards [5,6] and 28.2–45.5% in intensive care units [7,8] in Korea. In acute care hospitals, PI is most likely to occur within two weeks of admission [7], and in ICU, the risk of PI is even higher, with PI occurring mostly within 72 h of admission [9], and is reported to increase mortality by two to four times in elderly patients [8].

In addition to the economic losses associated with prolonged hospitalization and unnecessary medical expenditures for treatment and testing, PI outbreaks can lead to complications such as infection and even death if initial PI assessment and treatment are not provided appropriately [10]. Thus, PI prevention and management ensures patient safety and reduces medical costs, and many hospitals are actively performing preventive nursing care for PI, as PI-related items are included as evaluation data in the medical institution certification evaluation. PI is one of the activities for patient safety, and preventing and managing PI and evaluating the healing status is an important part of improving the quality of nursing care. The independent activities of nurses for preventing and managing PI should be based on the latest evidence [11].

The nurses’ role is to assess risk factors through daily direct nursing care by palpating the skin at PI-prone sites to identify the early stages of preventable PI and, if present, provide necessary interventions to prevent them worsening [12]. Accurately assessing PI and providing scientifically based nursing care is an important step [13]. In particular, nurses’ abilities to differentiate between pre and evident PI, incorporating the stages of PI during skin assessment, is critical [6].

The National Pressure Ulcer Advisory Panel (NPUAP) and the European Pressure Ulcer Advisory Panel (EPUAP) heavily informed the NPUAP-EPUAP International Classification System (2007) publication for PIs. After adopting the term “pressure injury” to include skin lesions without ulceration, it was renamed the National Pressure Injury Advisory Panel (NPIAP), which revised its classification in 2016 to include six stages: I, II, III, IV, unstageable PI, and, lastly, deep tissue pressure injury (DTPI) [14].

Conversely, incontinence-associated dermatitis (IAD) is an incontinence-related skin problem that results in inflammation and erythema due to prolonged exposure of the tissue to moisture from incontinence, enzyme irritation from feces, and perspiration [15]. IAD contributes to and aggravates the development of PI, and since 2000, PI and IAD have been evaluated separately [16]. The perineal area, where incontinence typically occurs, is anatomically close to the PI trigger site, and the high-risk populations for IAD and PI are similar [17]. Hence, differentiating between IAD and PI is challenging and can lead to inappropriate preventive care, treatment, reporting errors, and serious problems with nursing quality indicators [18]; hence, there is a need to improve nurses’ ability to differentiate between the two conditions to select the appropriate management approach based on the cause of the occurrence [19].

Nurses’ knowledge is important for improving PI nursing performance [20]. The knowledge-attitude-practice (K-A-P) model [21], which is utilized to explain human performance, shows that performance is influenced by knowledge and attitude. It has been reported that nurses’ PI nursing performance is low because nurses’ lack of PI nursing knowledge leads to poor understanding of PI-related practices [22], and lack of PI nursing knowledge and skeptical attitudes toward PI nursing negatively affect PI nursing performance [23]. It has also been shown that the higher the actual PI nursing knowledge, the higher the PI nursing performance [24].

Therefore, it is important for nurses to develop accurate knowledge of PI and IAD to accurately assess PI, detect it early, and provide efficient nursing care. However, previous studies in Korea have shown that nurses’ knowledge of PI is not high [25,26] and, in particular, their knowledge of PI status is relatively low compared to their knowledge of treatment and prevention of PI [27].

As PI assessment is the starting point for appropriate PI prevention nursing care, it is important to accurately determine nurses’ knowledge of PI assessment, which should be assessed by utilizing a visual differentiation ability test in conjunction with knowledge of their ability to accurately assess PI. In fact, there are previous studies on visual differentiation ability and knowledge to assess whether nurses can accurately assess PI according to the PI classification method [28,29,30,31]. Furthermore, a new theoretical framework related to pressure injury has been researched and published [32,33], and there are many papers that have studied the level of nurses’ knowledge of pressure ulcers based on this framework.

Therefore, it is necessary to investigate nurses’ visual differentiation ability of PI and to identify the factors that influence nurses’ visual differentiation ability of PI, such as knowledge and general characteristics of PI. Therefore, this study aimed to determine the knowledge and visual differentiation ability of clinical nurses regarding the six-stage PI classification system and IAD needing to be differentiated from PI, and to identify the factors that affect visual differentiation ability, so that it can be utilized as a basis for developing educational programs that can help efficient PI nursing practice in the future.

## 2. Research Methods

### 2.1. Research Design

This is a descriptive study aiming to determine the knowledge and visual differentiation ability of nurses in a hospital utilizing the PI classification system with IAD and to identify factors affecting their visual differentiation ability.

### 2.2. Participants

The participants were a convenience sample of 262 nurses who applied for continuing education in wound management and agreed to participate in the study at Chungbuk National University Hospital with about 1000 registered nurses. Finally, 248 questionnaires were analyzed after excluding 8 nurses who work independently as wound care specialist or outpatient clinic nurse and 6 whose questionnaires were not completed.

The minimum sample size for this study was calculated as 245 utilizing the G*Power 3.1.9.7 program (Düsseldorf University, Dusseldorf, Germany), a sample number calculation program, through regression analysis with a significance level of 0.05, a power of 0.95, a medium effect size of 0.10, and 9 predictors

### 2.3. Ethical Considerations 

This study was conducted after review by the Institutional Review Board of Chungbuk National University Hospital (IRB No. 2023-05-028-001). The purpose and contents of the study were explained to the participants, including voluntary participation and that the responses would not be utilized for any purpose other than the purpose of the study. Confidentiality of personal information was guaranteed, and that the participants could withdraw at any time if they wished. After confirming the participants’ willingness to participate in the study and obtaining their written consent, the study was conducted.

### 2.4. Measurement 

#### 2.4.1. General Characteristics

The general characteristics of the participants included demographic characteristics and PI nursing-related characteristics. Demographic characteristics included age, education, position, career length, and unit. PI nursing-related characteristics included frequency of caring for patients with PI or IAD experience, management of PI nursing care (PI risk assessment, wound assessment, dressing, and debridement), and participation in wound care education.

#### 2.4.2. PI Classification System and IAD Knowledge Test (PICS and IAD KT)

PI knowledge was measured with a 19-item instrument developed by Lee et al. [6] based on Kim’s [34] instrument. Using Kim’s [34] instrument for PI knowledge, content validity was verified by two adult nursing professors and four wound specialist nurses. Only questions with a content validity index (CVI) of 0.80 or higher were selected, and this was measured using the questionnaire tool, resulting in 19 questions. They consist of statements about the knowledge required to differentiate between PI assessment and IAD, including the pathogenesis, contributing factors, and wound characteristics of the PI classification system and IAD, which were reconstructed in six steps, with 1 point for correct answers and 0 points for incorrect answers. Scores range from 0–19, with higher scores indicating greater knowledge. The reliability at the time of development was Kuder–Richardson Formula 20 (K–R 20) = 0.76, and the validity was Content Validity Index (CVI) = 0.80. The reliability in this study was K–R 20 = 0.67.

#### 2.4.3. Visual Differentiation Ability Test of the PI Classification System and IAD (VDAT-PICS and IAD)

This is a tool developed by Lee et al. [6] to differentiate between the PI classification system and IAD and includes 21 photographs with information on the patient’s clinical condition, including medical diagnosis, patient mobility, bowel movement, presence or absence of percutaneous fluid infusion, and wound holding period. The 21 photographs consist of 16 pictures of each of the 6 stages of PI, 2 pictures of Blanching erythema, and 3 pictures of IAD, with 1 point for a correct answer and 0 points for an incorrect answer, and the score ranges from 0–21 points, with higher scores indicating higher visual differentiation ability. The reliability at the time of development was multi-rater kappa = 0.81 (*p* < 0.001), and the validity was CVI = 0.83. In this study, the reliability was K–R 20 = 0.82.

### 2.5. Data Collection Method

Data collection for this study was conducted on 22 June 2023 and 29 August 2023. This study measured PICS and IAD knowledge and visual differentiation ability utilizing questionnaires and photographs testing the participants. The questionnaire was completed directly by the participants and the PICS and IAD visual differentiation ability was measured via their assessments of the photographs and information provided in the printout utilized to complete the questionnaire. The time required to complete the questionnaire was approximately 15 min.

### 2.6. Data Analysis

Data were analyzed utilizing the SPSS WIN 27.0 program (IBM Corp, Armonk, NY, USA). Normality was verified using skewness and kurtosis. The general characteristics of the participants and the PICS and IAD KT and VDAT-PICS and IAD were analyzed utilizing frequencies, percentages, means, and standard deviations.
Differences in PICS and IAD KT and VDAT-PICS and IAD scores according to the general characteristics of the participants were tested via independent t-test and one-way ANOVA, and post hoc tests were analyzed via the Scheffé test. The correlation between PICS and IAD KT and VDAT-PICS and IAD was analyzed using Pearson correlation coefficients. Multiple linear regression analysis with enter method was conducted to identify factors affecting visual differentiation ability. The independent variables considered were PICS and IAD KT, frequency of caring for patients with PI or IAD and experience in PI management (assessing wound, dressing wound, and debridement). The categorical variable, frequency of caring for patients with PI or IAD, as well as the nominal variable of experience in PI management, were treated as dummy variables. To identify multicollinearity problems that can occur in multiple regression analysis, we used the variance inflation factor (VIF).

## 3. Results

### 3.1. General Characteristics 

The participants’ general characteristics are shown in Table 1. The mean age was 30.56 ± 8.03 years. About 84% of participants graduated with a bachelor’s degree or higher, and 89.1% were staff nurses. Half of the participants had clinical experience of <5 years, and the most common work departments were the internal medicine ward (32.3%), surgical ward (25.4%), and intensive care unit (21.8%). The highest frequency of care for patients with PI or IAD was often (36.3%), while the lowest frequency was never (10.1%). Regarding experience in caring for patients with PI, 81.5% had experience in performing PI risk assessment (Braden scale). A majority (62.5%) of participants had received wound care education at least once.

### 3.2. Descriptive Statistics of PICS and IAD Knowledge Test and Visual Differentiation Ability Test

As a result of measuring PICS and IAD KT, the score was 12.65 ± 2.90 out of 19 points, with a correct answer rate of 66.5%. When looking at the percentage of correct answers by item, “Moisture associated skin damage such as urinary and fecal incontinence is related to the development of pressure injuries.” was the highest at 96.8%, and “A stage III pressure injury involves the fat tissue and fascia.” was the lowest at 10.9%. “There is no necrotic tissue on wound bed in patients with incontinence-associated dermatitis.” was also among the least correct at 27.0% (Table 2).

When PI and IAD visual differentiation ability was measured by presenting photographs containing PI-related clinical information, the mean score was 11.43 ± 4.57 out of 21, with a correct response rate of 54.4%. The lowest response rate was 29.3% for unstageable PI photographs, followed by deep tissue pressure injury (DTPI) (47.3%) and stage III (55.6%). IAD, conversely, had the highest percentage of correct answers at 76.2% (Figure 1). When looking at nurses’ visual differentiation ability test errors, DTPI was most often confused with unstageable PI, followed by stage IV. Unstageable PI was the most difficult to distinguish from stage III, followed by DTPI and stage IV. Stage III was found to be confused with stage II and unstageable PI, in that order (Figure 1).

### 3.3. Differences between PICS and IAD Knowledge and Visual Differentiation Ability according to General Characteristics

The difference between PICS and IAD knowledge and visual differentiation ability according to the general characteristics of the participants is shown in Table 1. There were no significant differences in PICS and IAD KT according to general characteristics. Comparing the PICS and IAD visual differentiation ability according to the general characteristics of the participants, the results showed that frequency of caring for patients with PI or IAD (F = 3.15, *p* = 0.026) and thus experience in nursing patients with PI was significantly higher in wound assessment (t = −2.71, *p* = 0.007), dressing (t = 2.62, *p* = 0.009), and debridement (t = 2.54, *p* = 0.012). Hence, the more frequently PI care is performed and the higher the level of PI nursing performance, the higher the statistically significant visual differentiation ability test result. 

### 3.4. Factors Affecting PICS and IAD Knowledge and Visual Differentiation Ability

Although the r value was weak, there was a significant positive correlation (r = 0.252, *p* < 0.001) between PICS and IAD KT and VDAT-PICS and IAD. To identify factors affecting PICS and IAD visual differentiation ability, a regression model was built with significant PI management-related characteristics and PICS and IAD knowledge. Frequency of caring for patient with PI or IAD and experience in PI management, such as assessing wounds, dressing, and debridement, were entered as dummy variables, and PICS and IAD KT results were entered as continuous variables to build a regression model for PICS and IAD visual differentiation ability. The variance inflation factor (VIF) ranged from 1.03 to 3.08, all below the critical value of 10, indicating that multicollinearity is not a concern. The regression model of PICS and IAD visual differentiation ability was significant (F = 5.46, *p* < 0.001). The PICS and IAD knowledge, experience in debridement, and frequency of caring for patients with PI or IAD were significant factors. The explanatory power of the regression model constructed with these three variables for PICS and IAD visual differentiation ability was 13.7% (Table 3).

## 4. Discussion

The correct response rate for subjects in the PICS and IAD KT was 66.5% in our study. Comparing this rate with a previous overseas study [24] that examined not only knowledge but also attitude toward pressure injury and performance ability, it is challenging to make a direct comparison due to differences in study scopes. Nonetheless, it is worth noting that the current response rate is lower than that reported in the earlier study. It was low compared to the PI nursing knowledge level of 76% to 80%. This may be due to 50% of participants having less than 5 years of clinical experience and because the study was conducted within one hospital. Looking at individual items, among the 19 items, the correct answer rate was low for the knowledge questions about stage II and III pressure injury, and also for items related to the definition according to the characteristics of the wound area. Therefore, it is necessary to emphasize the precise definition of stages II and III, differentiation from stage IV, which includes fascia damage, a characteristic of the wound area, and stable management of eschar. 

To investigate the PICS and IAD visual differentiation ability, we evaluated the ability to classify PI and IAD utilizing photographs with clinical information, and the correct answer rate was 54.4%. In detail, the response rate for unstageable PI (29.3%), DTPI (47.3%), and stage III (55.6%) was low, but the visual differentiation ability for Blanching erythema, IAD, and stage II was over 70%. These results suggest that nurses need accurate education on unstageable PI, DTPI, and stage 3.

Looking at the incorrect answers of the items with low visual differentiation ability, DTPI was mainly confused with unstageable PI and stage IV, and unstageable PI was difficult to distinguish from stage III and DTPI. In the case of DTPI, nurses may be confused because there are reports that despite removing pressure and applying appropriate dressings in the clinical setting, there are cases of unstageable PI covered with necrotic tissue, after which the necrotic tissue is removed to expose the wound base, which is later found to be stage III and IV [23]. It is also believed that this may have caused nurses’ confusion because it is not uncommon to see multiple PI stages such as non-blanching erythema or blanching erythema, stage II, and unstageable PI on DTPI. Therefore, future PI- and IAD-related nurse continuing education programs should include precise definitions of unstageable PI and DTPI and clinical guidelines or changes in DTPI over time, and further clinical studies on changes in DTPI are needed. 

Regarding unstageable PI, due to the difficulty in distinguishing the black desiccated necrotic tissue covering the wound base in a planar view [23], it is likely that it was confused with DTPI, and the black desiccated tissue that had changed to yellow edible tissue on the screen was mistakenly identified as stage III. Similarly, stage III was incorrectly classified as unstageable PI because the adipose tissue at the base of the wound was confused with granulation tissue. Considering these results, it is necessary to provide practice-based education through simulation training or clinical practice utilizing videos on the characteristics of necrotic tissue and normal skin anatomy such as adipose tissue at different stages of skin damage. In addition, it is necessary to distinguish between blanching erythema and stage I, which is an important part of initial PI assessment and an important predictor of PI prevention, but in this study, there were difficulties in distinguishing between blanching erythema and stage I.

Factors that significantly affect the visual differentiation ability between PICS and IAD are the patient’s knowledge of PI, experience with debridement, and frequency of PI and IAD management. It was suggested that the higher the level of knowledge, the greater the experience with high-level surgery. PI nursing practice (debridement) with higher frequency significantly increases visual differentiation ability. In a previous study, visual differentiation ability was statistically significantly different depending on the medical staff’s experience in caring for patients with wounds, skin care, and PI prevention and treatment, and the higher the frequency of managing patients with PI or IAD, the higher the scores the PICS and IAD KT and VDAT-PICS and IAD [25], showing similar results to this study. This may have been influenced by the fact that clinical nurses have more opportunities to be exposed to work environments such as the management, prevention, and treatment of wounds and skin such as surgical sites and injection sites, even if they are not pressure injuries, and have increased their work proficiency through this. However, the explanatory power of the factors affecting nurses’ visual differentiation ability on PICS and IAD was low at 13.7%. Future researches are necessary for exploring other factors that affect nurses’ visual differentiation ability on PICS and IAD.

PI nursing knowledge is positively correlated with PI nursing attitudes, and PI nursing attitudes are positively correlated with PI nursing performance [25,35], while other studies have shown no correlation between PI nursing performance and PI nursing knowledge [36]. The results of these studies suggest many complex factors regarding the relationship between PI nursing knowledge and PI nursing practice. In general, it has been reported that even when knowledge is present, it is only likely to lead to practice when there is a high degree of certainty that the perceived knowledge is correct. Therefore, since PI nursing knowledge is acquired through relevant education, and PI nursing is performed based on PI nursing knowledge, a standardized PI nursing education system is needed so that PI nursing knowledge in clinical practice can be transferred to PI nursing performance [37]. In particular, a PI classification system education program should be provided to nurses to distinguish between IAD and PI, which are similar to PI and require differential diagnosis, and to increase visual differentiation ability for PI, and changes in theoretical knowledge and visual differentiation ability should be studied after education. Based on the results of this study, it is necessary to develop a PI education program that improves visual differentiation ability by increasing PICS and IAD knowledge to help nurses accurately assess PI and perform appropriate nursing care. In developing a PI education program, it is recommended that the contents of stage II, stage III, and IAD (which are items that clinical nurses have low knowledge of, as shown in the results of this study), and DTPI, unstageable PI, and stage III (for which they have difficulty in visual differentiation ability) should be strengthened and supplemented compared to existing PI programs.

The limitations of this study include the fact that it was conducted among clinical nurses in a single hospital and that the PICS and IAD KT tool utilized in the study was modified and supplemented from an existing tool, and it needs to be validated through future studies.

## 5. Conclusions

Accurate assessment and differentiation of PI from other skin lesions is crucial for the prevention and management of PI. In this study, the overall state of theoretical knowledge about PI and its risk factors was good, but the accuracy in the visual differentiation ability test for unstageable PI and DTPI was low, and confusion between stage I and blanching erythema and DTPI and unstageable PI was found. The visual differentiation ability was significantly influenced by knowledge of PICS and IAD, experience in debridement in PI patients, and frequency of caring for PI-related patients. Therefore, more PI nursing education experience is needed to improve nurses’ accurate PICS and IAD visual differentiation ability and improve the quality of nursing care to provide effective PI management. There is also a need for step-by-step training rather than one-time training on PICS and IAD. As a specific plan, education should be divided into an introductory course for nurses who manage PI and IAD patients intermittently, an advanced course for nurses who frequently manage patients or have extensive related education, and an advanced course for wound nurses. Excluding clinical experience and PI nursing performance, an accurate evaluation must be conducted by measuring the level of PI-related knowledge before training. In addition, we recommend a follow-up study to evaluate the effectiveness of continuous education by utilizing the PICS and IAD visual differentiation ability utilized in this study as a tool to measure the effectiveness of PI- and IAD-related education for nurses, measuring it before education and a certain period after education.

## Figures and Tables

**Figure 1 healthcare-12-00145-f001:**
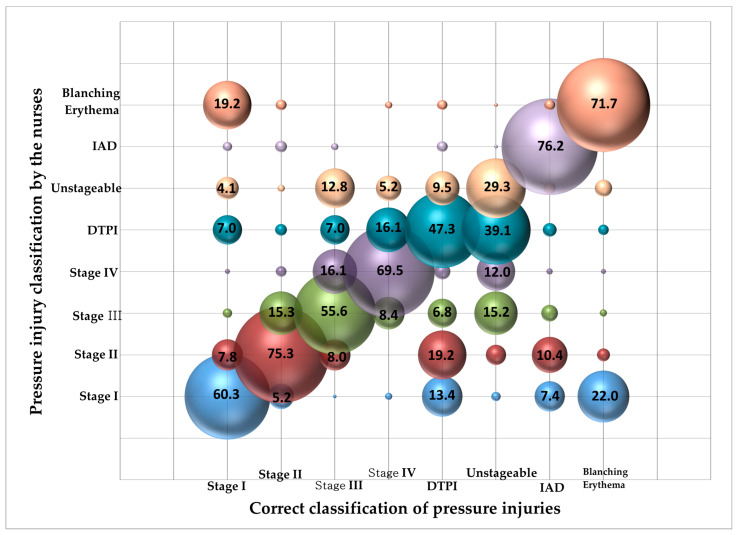
Correct answer rate on visual differentiation ability test by the nurses (N = 248). Notes: DTPI = deep tissue pressure injury; IAD = incontinence-associated dermatitis.

**Table 1 healthcare-12-00145-t001:** Comparison of the Pressure Injury Classification System and Incontinence-associated Dermatitis Knowledge Test and Visual Differentiation Ability Test (*n* = 248).

Characteristics	Categories	*n* (%)	PICS and IAD KT			VDAT-PICS and IAD		
			M ± SD	t or F	*p*	M ± SD	t or F	*p*
Age (year)	<30	156 (62.9)	12.76 ± 2.74	2.99	0.052	11.40 ± 4.50	0.91	0.404
	30–39	52 (21.0)	13.08 ± 2.65			12.04 ± 4.73		
	≥40	40 (16.1)	11.68 ± 3.61			10.75 ± 4.63		
		30.56 ± 8.03						
Education	Diploma	40 (16.1)	12.70 ± 3.49	3.61	0.059	11.53 ± 4.93	0.489	0.485
	≥Bachelor	208 (83.9)	12.64 ± 2.79			11.41 ± 4.50		
Position	Staff nurse	221 (89.1)	12.76 ± 2.81	2.23	0.109	11.63 ± 4.48	2.07	0.128
	Charge nurse	11 (4.4)	10.91 ± 3.42			9.36 ± 4.68		
	Manager	16 (6.5)	12.38 ± 3.54			10.06 ± 5.29		
Career length (year)	<5	124 (50.0)	12.75 ± 2.76	1.47	0.232	11.19 ± 4.45	1.35	0.261
	5–9	66 (26.6)	12.95 ± 2.48			12.21 ± 4.67		
	≥10	58 (23.4)	12.10 ± 3.55			11.05 ± 4.65		
Unit	Surgical	63 (25.4)	12.52 ± 2.84	2.00	0.094	10.49 ± 4.50	2.31	0.058
	Medical	80 (32.3)	12.88 ± 2.48			12.16 ± 4.51		
	ICU	54 (21.8)	12.41 ± 3.62			11.35 ± 5.11		
	ER	21 (8.5)	14.00 ± 2.26			13.10 ± 4.21		
	Others	30 (12.1)	11.83 ± 2.83			10.40 ± 3.49		
Frequency of caring for patients with PI or IAD	Never	25 (10.1)	12.52 ± 2.68	0.18	0.912	8.8 ± 4.13 ^a^	3.15	0.026 *
	Sometimes	52 (21.0)	12.90 ± 2.11			11.60 ± 4.07 ^b^		
	Frequently	90 (36.3)	12.63 ± 2.97			11.51 ± 4.38 ^b^		
	Usually	81 (32.7)	12.56 ± 3.34			12.01 ± 4.99 ^b^		
Experience in PI risk assessment	Yes	202 (81.5)	12.62 ± 2.98	−0.334	0.738	11.52 ± 4.52	0.70	0.483
	No	46 (18.5)	12.78 ± 2.55			11.00 ± 4.79		
Experience in assessing wounds	Yes	170 (68.5)	12.87 ± 2.98	1.75	0.082	11.95 ± 4.67	2.71	0.007 *
	No	78 (31.5)	12.18 ± 2.68			10.28 ± 4.12		
Experience in dressing wounds	Yes	163 (65.7)	12.60 ± 2.85	−0.39	0.697	11.97 ± 4.49	2.62	0.009 *
	No	85 (34.3)	12.75 ± 3.02			10.39 ± 4.56		
Experience in debridement	Yes	36 (14.5)	12.14 ± 2.82	−1.15	0.251	13.19 ± 3.77	2.54	0.012 *
	No	212 (85.5)	12.74 ± 2.91			11.13 ± 4.63		
Participation in wound care education	Never	93 (37.5)	12.77 ± 2.95	1.85	0.160	11.69 ± 4.93	1.16	0.317
	1–2	139 (56.0)	12.73 ± 2.84			11.44 ± 4.26		
	≥3	16 (6.5)	11.31 ± 3.01			9.81 ± 4.89		

Notes: PICS = Pressure Injury Classification System; IAD = Incontinence-Associated Dermatitis; KT = Knowledge Test; VDAT = Visual Differentiation Ability Test; a, b, = Scheffé test grouping; * *p* < 0.05.

**Table 2 healthcare-12-00145-t002:** Pressure Injury Classification System and Incontinence-Associated Dermatitis Knowledge Test (*n* = 248).

Items	*n* (%) or M ± SD
Moisture associated skin damage such as urinary and fecal incontinence is related to the development of pressure injuries.	240 (96.8)
Pressures and/or shearing force increase the risk for pressure injuries.	234 (94.4)
Secondary cutaneous infection such as fungal infection may easily develop in patients with incontinence-associated dermatitis.	233 (94.0)
A stage IV pressure injury is damage to muscle and bone.	226 (91.1)
The nose, ear, occiput, and malleolus do not have subcutaneous tissue and these injuries cannot become stage III.	221 (89.1)
Necrotic tissue, undermining, and tunneling may exist in stage III and stage IV	221 (89.1)
Deep tissue injury may further evolve and become covered by eschar. Evolution may be rapid exposing additional layers of tissue even with optimal treatment.	199 (80.2)
Stage I pressure injuries are defined as intact skin with non-blanchable erythema on bony prominence.	196 (79.0)
Deep tissue injury appears as an area of purple or maroon discoloration in intact skin or as a blood-filled blister.	194 (78.2)
Unstageable pressure injuries are wound whose bases are covered by dead tissues composed of slough and/or eschar.	178 (71.8)
Incidence of incontinence-associated dermatitis is higher in fecal incontinence than urinary incontinence.	175 (70.6)
It is possible to label the injury as Stage II when Stage III is healing with granulation tissue.	154 (62.1)
If there is perineal skin injury with erythema due to incontinence and no pressure, it is a pressure injury.	146 (58.9)
It is stage II if there is no bony prominence, but moisture associated skin damage with fecal incontinence.	113 (45.6)
Stable eschar on the heels serves as the body’s biological cover and should not be removed.	107 (43.1)
Stage II pressure injuries are intact skin with vesicles on pressure.	106 (42.7)
It is not a pressure injury if there are skin injuries with blanching erythema.	101 (40.7)
There is no necrotic tissue on wound bed in patients with incontinence-associated dermatitis.	67 (27.0)
A stage III pressure injury involves the fat tissue and fascia.	27 (10.9)
Total number of correct answers:	3138 (66.5)
Total Score	12.65 ± 2.90

**Table 3 healthcare-12-00145-t003:** Factors influencing Pressure Injury Classification System and Incontinence-Associated Dermatitis Visual Differentiation ability (*n* = 248).

Independent Variables		B	SE	Std. ß	t	*p*	VIF
Intercept		3.19	1.49		2.14	0.034	
Knowledge of PICS and IAD		0.41	0.10	0.26	4.25	<0.001	1.03
Experience in assessing wounds †		0.38	0.67	0.04	0.56	0.577	1.31
Experience in dressing wounds †		0.87	0.66	0.09	1.33	0.186	1.31
Experience in debridement †		1.65	0.83	0.13	1.99	0.047	1.15
Frequency of caring for patients with PI or IAD †	Sometimes	2.10	1.06	0.19	1.98	0.049	2.50
	Frequently	2.06	1.00	0.22	2.06	0.040	3.08
	Usually	2.56	1.01	0.26	2.53	0.012	3.01
Model					5.46	<0.001	
Durbin–Watson = 1.998, R^2^ = 0.137, Adj R^2^ = 0.112, F = 5.46, *p* < 0.001							

Notes: PICS = Pressure Injury Classification System; IAD = incontinence-associated dermatitis; VIF = variance expansion factor; † dummy variables.

## Data Availability

Data are contained within the article.

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
