# Peer review of "Knowledge and Visual Differentiation Ability of the Pressure Injury Classification System and Incontinence-Associated Dermatitis among Hospital Nurses: A Descriptive Study"

_healthcare, 2024, doi:10.3390/healthcare12020145_

Round 1

Reviewer 1 Report

Comments and Suggestions for Authors

Interesting study about nurses' knowledge and visual differentiation ability of PI assessment.

I have several notes for the authors in order to increase its quality.

GENERAL COMMENTS :

Please uniform along the text the reference to the instruments PICS & IAD KT and VDAT-PICS & IAD.

I’m not native English. In general, the paper seems well written. However, as I will mention in my comments, I found that some sentences were too long. I suggest you divide some of them in order to make your text easier to understand.

INTRODUCTION

One of the aims of the work is “to identify the factors that affect them [knowledge and visual differentiation ability]”, but only a regression model was built for visual differentiation ability.  Shouldn't you also have built a model for knowledge?

RESEARCH METHODS

Section 2.2:

- Clarify whether 262 is the size of the population or the sample. If it is the sample, what is the size of the population being studied?

- How was the sample selected? The abstract states that it was a convenience sample, but nothing is said about the research design or participants. Also, some more detail needs to be given.

- It is mentioned that “8 nurses who met the exclusion criteria”, but what was these criteria? Or what was the inclusion criteria?

- In short, you need to give more details about how the sample was selected/obtained.

Section 2.4.2: It is written “PI knowledge was measured with a 19-item instrument developed by Lee et al. [6]”, but in Lee et al. there is no reference to this instrument.

Sections 2.4.2 and 2.4.3: If both instruments, PICS & IAD KT and VDAT-PICS & IAD, were developed by Lee et al. [6], having both a dichotomic scores in each item, why those authors used a different measure to evaluate the reliability? (Kuder and Richardson Formula 20 and Multi-rater kappa).

Section 2.5 (Data Collection Method): How the participants were contacted to take part in the study. How was the information collected (on paper, in person, online, ...)?

Section 2.5 (Data Analysis):

- Correct the section numbering. It should be 2.6.

- There is no need, and is unusual, to use a bullet list to describe the data analysis. Just remove the bullets.

- Have you validated the assumptions of the tests used?

- What type of regression model was used? Linear? Nonlinear? Clarify, please.

- What was the method used to select the variables in multiple regression? And what were the initial factors considered in this regression? Part of this information is presented in section 3.4 instead of being presented here.

- In section 3.4 you start by presenting a correlation result (and also on table 3), but is this section 2.6 is missing the reference to this analysis and the correlation coefficient used.

RESULTS

Section 3.1:

- Review the text of the general characteristics. It repeats the information that is available in Table 1. You just should highlight some of that information.

- Half of the participants have clinical experience < 5 years, and the majority (62.5%) had received wound care education. Among this “majority” most of them have <5 years or more of clinical experience? I.e., it is more likely that participants with more years of clinical experience have wound care education or not?

- Table 2: For ease of reading, I suggest presenting the items in order (ascending or descending) of frequency. In the last line, instead of total I think it should be total score.

Section 3.2, Figure 1: Improve the figure. I suggest reducing the size of the bubbles to reduce overlap and improve the readability of the results. Improve the text size in the figure (is not in accordance with the text). Remove the horizontal line above the x text values.

Section 3.3: Differences between PICS & IAD knowledge and visual differentiation ability according to general characteristics: there is no text about PICS & IAD KT. Even if there are no significant differences, this result must be reported.

Section 3.4:

- 1st sentence: It should be referred that the correlation is weak. And this first sentence should be in an isolated paragraph.

- Table 3: all the information they present in table 3 is available in the first sentence of this section. Therefore, the table is unnecessary. As an alternative to the table, I suggest you present a scatter diagram as it allows you to see if by any chance there is some kind of non-linear relationship between the scores of the two tests.

- What do you mean by "input variable selection method"? There are several methods to select the variables. Which one was used? Maybe you are referring to the SPSS notation. I suggest to check SPSS help to find what method was used.

- I think that the word “among” (line 234) should be eliminated, otherwise the sentence does not make sense.

- If you used a linear regression model, have you considered the use of a generalized linear model? Have you compared the coefficients of the several simple regression models with the coefficients of the multiple regression model to check if the is no big differences? Have you checked for any confounding variables?

- The R2 is too low. Something should be said about it (what is the point of a model with such a low R2?).

- Line 230-231: If I understand well, PICS & IAD knowledge test is just one variable. Therefore, it must be corrected to “PICS & IAD knowledge test was entered as continuous variable”

DISCUSSION

Please review this section. Do not repeat the presentation of results. In this section you should interpret your results, what they mean, present their implications, and connect them to the related research. Sometimes this is not done. I will just highlight some sentences and paragraphs, but a general revision in needed. For instance:

- 1st sentence: It does not make sense to write “which is difficult to compare directly due to the different subjects and tools” and then refer that is “lower than previous studies [23,27] in which the nurses' PI nursing knowledge level was over 76%–80%”, since you are comparing!

- Lines 248-252, 257-260: Please, avoid repeat the presentation of the results. The focus should be in discussing them.

- Line 285-286: It is not clear whether only the results presented in this paragraph are similar to those of previous studies, or whether the results presented in the previous paragraphs are also similar to those of these studies.

- Lines 287-291: the sentence is to long!

- Line 295: are you sure that the reference to [27] is correct?

- Lines 300-320: where is the link to your findings?

- Line 324: remove the word repeated.

CONCLUSION:

- Some sentences are too long, for instance, lines 332-336.

- Lines 337-340: the same can be written with a sorter sentence. With the creation of introductory and advanced courses, by experience and knowledge of the nurses, aren’t you adjusting the level of difficulty to match the knowledge level of the participants

Author Response

Dear Reviewer 1.

Thank you very much for waiting for a long time for my answer.

Thank you.

Reviewer 2 Report

Comments and Suggestions for Authors

Dear authors, thank you for provide me the opportunity to review this manuscript who falls into my area of expertise.

The manuscript, in general, is well written and sounds well from the methodological point.

I only have some comments to improve your manuscript, if you consider that:

- The topic of your manuscript was researched since 2000's by Defloor team and colleagues of EPUAP. Since then was stablished that there is a difficulty in differentiate between PI categories and, also, IAD. And was stated that learning on classification should be improved. This paper reflects that more than 20 years later the problem remains.

- My main comments is regarding the references used (or not used) on the introduction section and discussion. Since the first papers and statements documents written by the EPUAP Society members and others, about pressure injuries and IAD differentiation, new models and theories have emerged on the literature explaining how a pressure injury or a IAD is developed and how to differentiate (Ethically, I must to state that in some of the cited below, I'm co-author and is not my intention that you must to reference me. This is only to explain that there is a growing field of new knowledge around pressure injuries and other related lesions and this must to be stated, because this could have implications on knowledge and attitudes).

For instance, you cited a paper By Defloor et al, 2006 but there is another with bigger sample, in the same topic, on 2007 who was not mentioned (DOI: 10.1111/j.1365-2648.2007.04474.x).

Other papers where new frameworks or models were developed include:

- A new pressure ulcer conceptual model (doi: 10.1111/jan.12405)

- A new theoretical model for the development of pressure ulcers and other dependence-related lesions (DOI: 10.1111/jnu.12051)

On the other hand, there are other papers based on knowledge of pressure ulcers using different instruments who could be used to compare with yours, as most of those used here are Korean-based and likely by the same research group (self-citation).

- What is the difference, despite the sample size and that is written in Korean, of reference 26, and this manuscript? are different hospitals or settings?

- Regarding the results, the text about correlation between tests and table 3 is redundant as are showing the same information and there is not any new information, then, likely the table could be deleted.

Author Response

Dear Reviewer 2.

Thank you very much for waiting for a long time for my answer.

Thank you.

Round 2

Reviewer 1 Report

Comments and Suggestions for Authors

Dear authors,

Thanks for the revised version of the manuscript and for attending most of my comments. However, I still have some comments and suggestions.

1) A limitation you point out is the need for PICS & IAD KT tool to be validated. There are several validated questionnaires to assess nurses’ knowledge about wounds, and they have even been translated into several languages. Why didn’t you use one of those questionnaires? What is the advantage of PICS & IAD KT?

Section 2.4.2: PI classification system and IAD knowledge test (PICS & IAD KT)

2) According to Lee et al. [26], Lee et al. (2011) [6] developed a visual tool with 21 photos, whereas Kim (2003) developed a multi-item knowledge instrument of which Lee et al. (2013) [26] considered only the 19 items with CVI>=0.8. In other words, Lee et al. [6] and Kim developed different instruments, or am I wrong? Please clarify this in the first sentence of this section.

Section 2.6: Data analysis

3) You have to mention how you validated the assumptions. For instance, you can include your response sentence: “Normality was tested using skewness and kurtosis”

4) You have to mention which correlation coefficient was used. It seems to me it was Pearson correlation coefficient.

5) Regression:

5.1) “enter [and not entry] method” is not a variable selection method. This is a term used in SPSS, not in statistics. The “enter method” in SPSS means that all independent variables will be included in the model, and none will be out of your model (https://www.ibm.com/docs/en/spss-statistics/29.0.0?topic=regression-linear-variable-selection-methods). However, according to what you have written in the results section, it seems that the model you present in Table 3 do not have all the independent variables you have considered, or am I wrong? In that case, you have used a method (stepwise, backward or forward) to select the independent variables in your final model. Please clarify!

5.2) Have you check for multicollinearity?

5.3) In the data analysis section, and not in the results section, you should refer what were the initial independent variables considered in your model. Write in here what variables were included as numeric and what variables were included as dummy variables. If you used a variable selection method, mention in this section what was the method.

Section 3: Results:

6) I don't think that “subject” is the best work. Usually it is used participants, or nurses, or respondents.

7) Lines 180-181: The sentence needs to be rephrased. here's a suggestion: “Half of the participants have clinical of <5 years, and the most common work departments were internal medicine ward (32.3%), surgical ward (25.4%), and intensive care unit (21.8%)”.

8) Lines 182-183: If you report all the categories and values (of the variable frequency of caring) that you are presenting in the table, then the presentation of that information in the table no longer makes sense. Therefore, please, only report the main results in the text (it can be the most frequent(s) or the least frequent(s)).

9) Figure 1: I think the bubbles could be a little smaller to avoid so much overlap. I think you could use a slightly lower ratio. In stage 4 (x-axis) and stage III (y-axis) the value in the bubble is missing.

10) Line 233: “The input variable selection method was utilized to build the model”. What do you mean by “input variable selection method”, because input is not a method.

11) Sorry, I don’t understand your response 26: "Inputting the variables that emerged as significant in Table 2 and correlation into the regression model took confounding factors into consideration".

Section 4: discussion

12) Lines 242-245: Please rephrase this long sentence. It is really difficult to understand. Suggestion: 1st sentence: present your response rate; 2nd sentence: comment on whether tour value it is higher or lower than other studies, and add the reference to that studies; 3rd sentence: state that, however, these rates cannot be directly compared since ... [identify the differences in the studies].

13) Line 246-247: “it can be assumed” is too strong and can not be correct. I suggest that instead of “This can be assumed to be because 50% of the subjects had less than 5 years…” it should be “This may be due to 50% of participants having less than 5 years …”

14) Line 250: Instead of “and in addition, the correct answer rate was low for items related to the definition according…” it should be “and also for items related to the definition according…”

15) Line 252-253: I think that instead of “damage, which is a characteristic…” it can be “damage, a characteristic…”

Author Response

Dear reviewer 

Thank you for your thoughtful review.

Round 3

Reviewer 1 Report

Comments and Suggestions for Authors

Dear authors,

Thank you for the revised version of the manuscript and for responding to most of my comments.

I still have a few observations, many of which are English suggestions.

1) line 111: Remove the “.” before “The participants”.

2) line 171: I know it was my suggestion, but instead of “tested” it should be “analyzed”, “verified” or “evaluated”, as “tested” means that you carried out a test which was not the case.

3) lines 180-188: Please rephrase. You should not begin a sentence with the word And. VIF=1/TOL, so it is redundant to present both of these measures. The most common is VIF. Suggestion: "The independent variables considered were PICS & IAD KT, frequency of caring patients with PI or IAD and experience in PI management (assessing wound, dressing wound, and debridement). The categorical variable, frequency of caring patients with PI or IAD, as well as the nominal variables experience in PI management, were treated as dummy variables. To identify multicollinearity problems that can occur in multiple regression analysis, it was used the variance inflation factor (VIF)."

4) lines 252-256: rephrase, please:

4.1) lines 252-253: Remove the sentence ".The enter method was utilized to build the model".  This has already been written in section 2.6.

4.2) lines 253-254: As I wrote above, VIF=1/TOL, so remove the sentence " Multicollinearity was assessed, and the tolerance limit (TOL) ranged from .32 to .97, well above the acceptable threshold of 0.1".

4.3) lines 255-256: Instead of "...indicating that the variables were independent." (which is not true, because they can be weakly correlated) write "...indicating that multicollinearity is not a concern."

5) lines 256-259: there are some mistakes in these sentences that require correction. You open a "(" but don't close it ")". Furthermore, some variables mentioned are not significant factors. Suggestion: "The regression model of PICS & IAD visual differentiation ability was significant (F=5.46, p<.001). The PICS & IAD knowledge, experience in debridement, and frequency of caring patients with PI or IAD were significant factors."

6) Table 3: some changes are needed:

6.1) The symbol † is missing in the 3 variables “Experience in…”

6.2) Remove the TOL column.

6.3) Present the p-value in a separate column from t.

6.4) The values presented in the various measures must correspond to the complete model presented in Table 3, and not the model with only the significant variables.

6.5) Footnote: remove "TOL = Tolerance limit."

7) line 267: instead of “was 66.5%, as indicated in the previous study [23]” it should be “was 66.5% in our study”.

Keep up the good work!

Author Response

Dear Reviewer.

Thank you for your thoughtful review.
